# Recovering Secondary REE Value from Spent Oil Refinery Catalysts Using Biogenic Organic Acids

**Melisa Pramesti Dewi [1], Himawan Tri Bayu Murti Petrus [2,3] and Naoko Okibe [1,\*]**

[1] Department of Earth Resources Engineering, Kyushu University, Fukuoka 819-0395, Japan; melisapdewi@gmail.com

[2] Department of Chemical Engineering, Universitas Gadjah Mada, Yogyakarta 55281, Indonesia; bayu.petrus@ugm.ac.id

[3] Unconventional Georesources Research Center, Faculty of Engineering, Universitas Gadjah Mada, Yogyakarta 55281, Indonesia

\* Correspondence: okibe@mine.kyushu-u.ac.jp; Tel.: +81-92-802-3312

**Abstract:** Spent catalysts produced by oil refinery industries are regarded as an important secondary source for valuable metals. In particular, spent fluid catalytic cracking (FCC) catalysts represent a potential source for rare earth elements (REEs). This study aimed to exploit the leachability of spent FCC catalysts as a secondary source for La, by using an alternative organic acid lixiviant produced under optimized fungal fermentation conditions. The first chemical leaching tests revealed that citric acid (>100 mM) is a comparable alternative lixiviant to conventional inorganic acids (1 M) and that the La dissolution behavior changed significantly with different types of organic acids. The initial fungal fermentation conditions (e.g., inoculum level, substrate concentration, pH) largely affected the resultant biogenic acid composition, and its manipulation was possible in order to almost solely ferment citric acid (~130 mM) while controlling the production of unwanted oxalic acid. The performance of actual biogenic acids (direct use of cell-free spent media) and artificially reconstituted biogenic acids (a mixture of chemical reagents) was nearly identical, achieving a final La dissolution of ~74% at a pulp density of 5%. Overall, the microbiological fermentation of organic acids could become a promising approach to supply an efficient and environmentally benign alternative lixiviant for REE scavenging from spent FCC catalyst wastes.

**Keywords:** spent fluid catalytic cracking (FCC) catalyst; rare earth element (REE); lanthanum (La); citric acid; fermentation; fungi; *Aspergillus niger*

## 1. Introduction

Oil refinery industries rely extensively on solid catalysts to facilitate process efficiency. Such hydroprocessing (hydrotreating/hydrocracking) and fluid catalytic cracking (FCC) catalysts are subject to deactivation due to coke, sintering and contamination. Still, they are generally reused multiple times through regeneration treatments before coming to the end of the cycle [1]. Due to the presence of hazardous heavy metals, spent catalysts can cause serious environmental pollutions if not properly handled, and are classified as the most dangerous wastes to be generated in oil refineries [2].

FCC catalysts are produced at ~840,000 metric tons per year and account for a major application of rare earth elements (REEs) [3]. REE ions, most classically La and Ce, migrate into the zeolite structure to improve the catalyst reactivity, selectivity and thermal stability by reducing dealumination and loss of crystallinity [3,4]. The use of La, for example, enables highly energy-efficient petroleum cracking, and nearly half of the global consumption of $La_2O_3$ is dedicated to the production of FCC catalysts [5]. Due to their highly uneven geographical distribution, together with geopolitical issues

in corresponding countries, the recovery of REEs from secondary sources is critical to realizing their sustainable supply.

Therefore, despite the environmental concerns, the spent catalysts are regarded as an essential secondary source for valuable metals, and many attempts have been made to extract such metal values via pyrometallurgical or hydrometallurgical methodologies (e.g., smelting, acid leaching, caustic leaching; [6,7]).

In addition to the common inorganic acid lixiviant used for hydrometallurgical treatment, studies have been conducted in order to utilize organic acids to extract metal values from both natural ores [8] as well as waste materials [2,9]. A variety of organic acids (e.g., citric acid, oxalic acid, gluconic acid, malic acid, succinic acid) is known to be fermented by filamentous fungi, out of sugar-containing wastes or the degradation products of lignocellulosic biomass wastes [10]. The type and amount of organic acids produced by fungal fermentation are determined by factors such as the carbon source, nitrogen contents, phosphate, heavy metals and other trace elements, and physicochemical conditions for fungal growth [2,11]. *Aspergillus* spp. (especially *A. niger*) and *Penicillium* spp. are especially well-studied fungal genera for organic acid production [10,12]. Among these organic acids, citric acid is regarded as being the most important due to its low toxicity, with a broad applicability in the pharmaceutical, food, cosmetics and toiletry industries [13]. The commercial production of citric acid mainly employs submerged fermentation using *A. niger* strains [13]. Finding the utility of such organic acids in the metallurgical industry would shed further light on their applicability.

Metal leaching with fungal metabolites is based on the combination of the following reactions:

Acid leaching:

$$MeO + 2H^+ \leftrightarrow Me^{2+} + H_2O \tag{1}$$

Metal-organic acid complexation:

$$Me^{2+} + H_2\text{-}A \leftrightarrow Me\text{-}A + 2H^+ \tag{2}$$

The metals could be recovered from organic leachates via solvent extraction or sulfide precipitation [14]. Aung and Ting [15] leached the FCC catalyst (at 1–12% pulp densities, $D_{mean}$ = 94 μm) using biogenic acids fermented from sucrose by *A. niger* (composed of 57 mM citric acid, 48 mM gluconic acid and 7 mM oxalic acid); the results were compared with chemical leaching tests using inorganic (sulfuric and nitric acids) and organic acids (citric, oxalic and gluconic acids). Cell-free biogenic acids generally resulted in a greater metal dissolution (64% Sb, 9% Ni, 23% Fe, 30% Al and 36% V at a 1% pulp density in 50 d) than the artificially reconstituted organic acid mixture. Reed et al. [16] observed a maximum REE leaching of 49% from the spent FCC catalyst ($D_{mean}$ = 80–90 μm) using cell-free spent media of the gluconic acid-producing bacterium *Gluconobacter oxydans*, with the preferential recovery of La over Ce. Biogenic acids containing 10–15 mM gluconic acid were more effective than abiotic gluconic acid of higher concentrations, suggesting the importance of other cell exudate components. Mouna and Baral [17] leached La from the FCC catalyst using *A. niger* and compared the results with chemical leaching tests using different inorganic and organic acids. A one-step bioleaching at 1, 3 and 5% pulp densities yielded a 63%, 52% and 33% La dissolution, respectively. A two-step bioleaching using a cell-free spent medium was significantly less effective. A chemical leaching test using sulfuric and nitric acid showed a 38% efficiency, whereas oxalic acid resulted in a 5% La dissolution. Using hydrochloric, citric and gluconic acids resulted in a 68%, 65% and 64% La dissolution, respectively, which was nearly equivalent to that offered by the one-step bioleaching. Despite the importance of REE recycling from spent catalysts, studies focusing on this topic are still scarce. In particular, the potential utility of bioprocessing needs to be exploited within sustainable technologies.

In countries such as Indonesia, where both the oil and agricultural industries play essential economic roles, scavenging secondary metal values out of oil refinery wastes via the fermentation of agricultural wastes would bring about cross-industrial benefits from environmental as well as economic aspects. The potential application of fungi for metal leaching from solid materials was proposed

by Burgstaller and Schinner [18], wherein the effectiveness of fungal leaching agents (organic acids) in forming metal complexes and the particular niches of such heterotrophic microorganisms were mentioned (where cheap organic wastes, such as permeate or molasses, were available). This study thus aimed to exploit the recyclability of spent FCC catalyst waste as a secondary source of La, one of the critical REEs, by using an alternative organic acid lixiviant produced under optimized fungal fermentation conditions.

## 2. Results and Discussion

### 2.1. Characterization of Spent FCC Catalyst

The spent FCC catalyst was comprised of spherical particles with an average diameter of 80 μm. Table 1 shows its metal composition, estimated from a sequential acid digestion followed by ICP-OES (Inductively coupled plasma-optical emission spectrometry) measurements. La was mostly dissolved in Fraction 1 but not in Fraction 2, whereas most of Ni was dissolved in Fraction 2 together with Si. This indicates that Ni was associated with silicate mineral but that La was not. About 20% of Al being dissolved in Fraction 1 and the rest in Fraction 2 suggests that Al exists in the spent FCC catalyst as both $Al_2O_3$ (dissolved in Fraction 1) and zeolite-Y $((Na_2,Ca,Mg)_{3.5}[Al_7Si_{17}O_{48}]\cdot32H_2O$; dissolved in Fraction 2, so mostly as Fe). Since the components solubilized in Fraction 3 are generally sulfides (Section 3.2.2.), Table 1 suggests that La is partially present in the form of sulfide in the spent FCC catalysts.

**Table 1.** Elemental composition of the spent FCC catalyst determined by a sequential acid digestion followed by ICP-OES analyses.

| | wt. % * | | | |
|:---:|:---:|:---:|:---:|:---:|
| **Element** | **Fraction 1** | **Fraction 2** | **Fraction 3** | **Total** |
| Al | 4.20 | 13.79 | 0.03 | 18 |
| Si | 0.15 | 17.70 | - | 18 |
| La | 1.64 | - | 0.24 | 1.9 |
| Ni | 0.05 | 1.13 | - | 1.2 |
| Fe | 0.08 | 0.22 | - | 0.3 |

* Results obtained are the average of duplicate determinations.

The typical chemical and structural compositions of FCC catalysts were precisely described by Vogt & Weckhuysen [3]. Zeolite (usually stabilized zeolite-Y) is the main active component of the FCC catalyst. Zeolite functions as an internal porous structure with acid sites, which can convert larger molecules to the desired gasoline range molecules. Alumina and silica components are used as a meso- and macro-porous matrix to precrack larger molecules, as well as to bind the system. Clay is added as a filler, as well as for heat-capacity reasons. Additional components may comprise specific traps for contaminant metals such as Ni and V from crude oil. These components are typically mixed in an aqueous slurry and then spray-dried to form spherical particles that can be refluidized in the regenerator [3]. The XRD patterns of the catalyst sample used in this study before and after two types of HF treatments are shown in Figure 1. The main mineral components of the spent FCC catalysts were zeolite-Y and $Al_2O_3$ (Figure 1a). Two different HF treatments were applied in order to partially digest such a main catalyst matrix for the visualization of minor La- or Ni-containing minerals. After a 23% HF treatment at room temperature, the XRD peaks of the zeolite-Y matrix diminished, while $Ni_2SiO_4$ peaks became visible (Figure 1b). As was suggested from the sequential acid digestion results, Ni originating from the feed oil was trapped and mineralized to form $Ni_2SiO_4$. Since Ni contaminants cause catalyst deactivation, the catalyst regeneration process is set to stabilize Ni in the silicate component under high-temperature conditions (personal communication; PT Pertamina (Persero), Indonesia). The microwave-assisted 23% HF treatment dissolved not only zeolite-Y but also $Al_2O_3$, enabling peaks of $AlF_3$ and synthetic $LaF_3$ to emerge instead (Figure 1c). The existences of $AlF_3$

and synthetic LaF$_3$ were the results of the reactions in Equations (3) and (4). According to the catalyst manufacturer, fresh FCC catalysts originally contained La in the form of La$^{3+}$ in the zeolite-Y matrix, which then produced La$_2$O$_3$ after being used in the FCC process.

$$Al_2O_3 + 6HF \rightarrow 2AlF_3 + 3H_2O \tag{3}$$

$$La_2O_3 + 6HF \rightarrow 2LaF_3 + 3H_2O \tag{4}$$

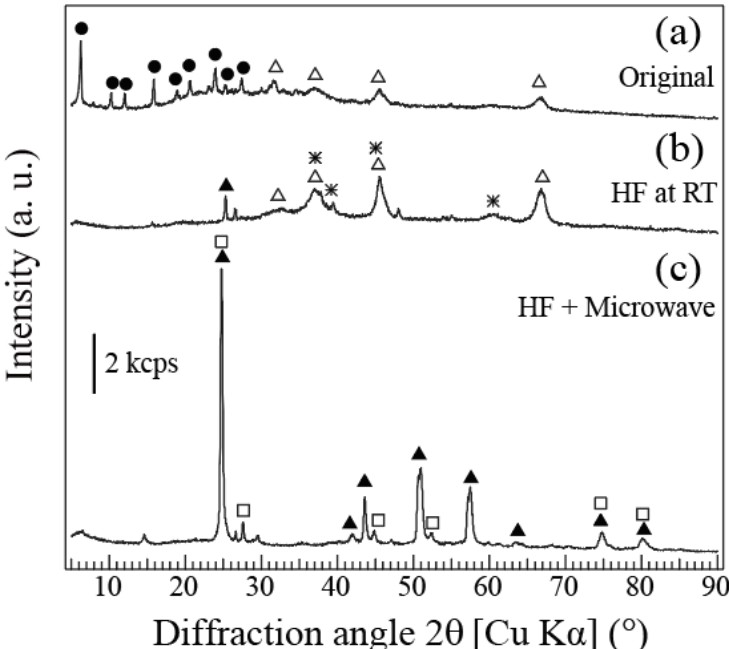

**Figure 1.** XRD patterns of spent FCC catalysts (**a**) before and after a 23% HF treatment under (**b**) atmospheric or (**c**) microwave conditions. Conditions: zeolite-Y (●; PDF No. 01-073-7918), Al$_2$O$_3$ (△; PDF No. 01-074-2206), AlF$_3$ (▲; PDF No. 01-072-1117), Ni$_2$SiO$_4$ (*; PDF No. 01-083-2071), synthetic LaF$_3$ (□; PDF No. 00-032-0483).

### 2.2. Chemical Leaching of Spent FCC Catalysts

First, the effect of citric acid was compared to a regular inorganic acid lixiviant such as HCl and H$_2$SO$_4$ at 1 M. The initial speed of the La dissolution was greater with inorganic acids. Still, the final La dissolution was nearly comparable in all cases, reaching ~80% at 70 h (Figure 2a). Nickel was hardly dissolved (Figure 2c), and Al was dissolved to <20% (Figure 2b). According to Table 1, about 78% of Al in the spent FCC catalyst originated from zeolite and 23% from Al$_2$O$_3$. Since the latter mineral was more soluble than the former, the Al dissolved in Figure 2b was likely leached from Al$_2$O$_3$. Using the same inorganic acids at a higher concentration of 2 M resulted in a nearly identical dissolution behavior of metals (Figure S1). At the same concentration, citric acid can become a comparable alternative to inorganic acids, while keeping a higher pH level (Figure 2d). This implies that, in addition to acidolysis, there indeed exists another leaching mechanism with citric acid, that is, a complexation of metals with organic functional groups.

The citric acid molecule possesses three carboxylate groups, each of which holds four free electron pairs. These carboxylate groups are known to form stable complexes with metal ions, and La$^{3+}$ was also shown to form a nine-fold coordination with oxygen atoms (7 from citric acid and 2 from water molecules [19]). Each citric acid molecule binds with three La$^{3+}$ ions, while one La$^{3+}$ ion binds with three molecules of citric acid, eventually forming La-citrate complexes with a molar ratio of 1:1 [19].

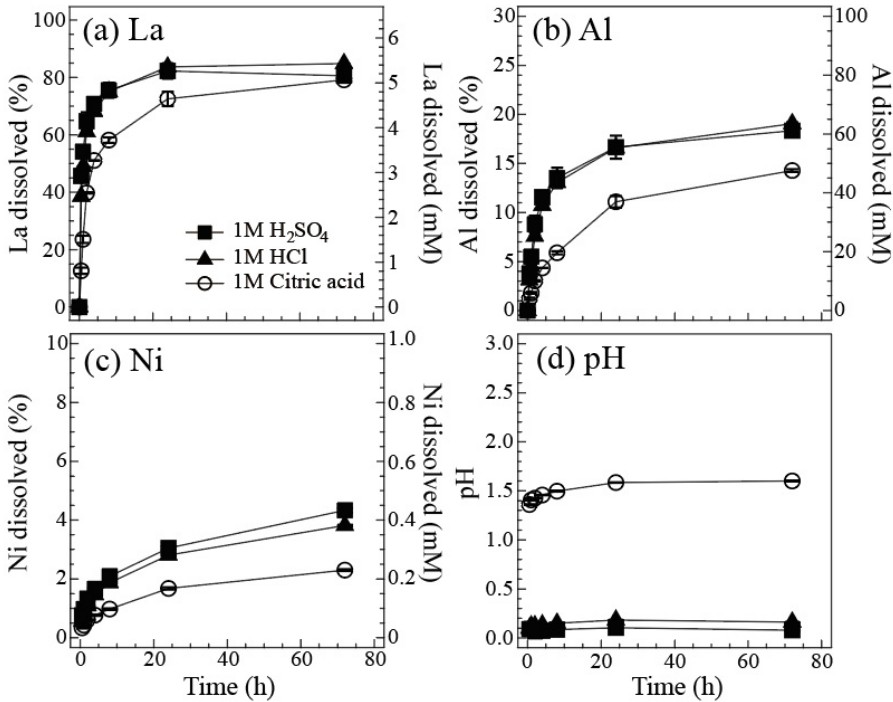

**Figure 2.** Changes in the total soluble concentrations of (**a**) La, (**b**) Al and (**c**) Ni, and in (**d**) the pH during the chemical leaching of the spent FCC catalysts using 1 M $H_2SO_4$ (■), 1 M HCl (▲) or 1 M citric acid (○).

Since the effectiveness of citric acid was evident, the next chemical leaching tests compared the effect of different organic acids at a fixed concentration of 50 mM. The final La dissolution at 72 h differed largely between the acids, resulting in 68% 40%, 26% and 0.2% by citric, lactic, succinic and oxalic acid, respectively (Figure 3a). The dissolution of Al was < 12% under all conditions (Figure 3b). The surface of the spent FCC catalysts after the leaching test is shown in Figure 4. The stability constants (log β; a measure of the strength of the interaction between a metal ion and a ligand in forming a metal complex) of aqueous La-organic acid complexes are, in descending order, 9.1 for citric acid [20], 6 for oxalic acid [21,22], 4.3 for succinic acid [20] and 2.9 for lactic acid [22]. The number of carboxyl groups for oxalic, succinic and lactic acids are 2, 2 and 1, respectively. Nonetheless, the La dissolution with oxalic acid turned out to only be negligible (Figure 3a). This observation can be theoretically explained by the low solubility product of La-oxalate ($La_2(C_2O_4)_3$; $\log Ksp = -29.15$; [20]), with which $La^{3+}$ ions leached from the catalyst would immediately precipitate at 50 mM oxalic acid. SEM images, shown in Figure 4, revealed the formation of distinctive precipitates on the surface of the spent FCC catalysts after the leaching test, especially in the case of oxalic acid (Figure 4e) and succinic acid (Figure 4d). The initial pH value in each organic solution was consistent with the theoretical calculation based on the acidity constant (pKa = 1.3 (oxalic acid), 3.1 (citric acid), 3.9 (lactic acid), 4.2 (succinic acid)). The pH fluctuation in organic acid leaching solutions can be affected by; (i) the proton-consuming acid leaching reaction (Equation (1)), (ii) the deprotonating complexation reaction between carboxyl groups and metals (Equation (2)) and (iii) the natural buffering capacity of organic acids. Compared to other organic acids, a pH rise was noticeable in oxalic acid solutions, while the La dissolution was inhibited (Figure 3c). This observation indeed suggests the immediate precipitation of La-oxalate and the loss of the buffering capacity of oxalic acid.

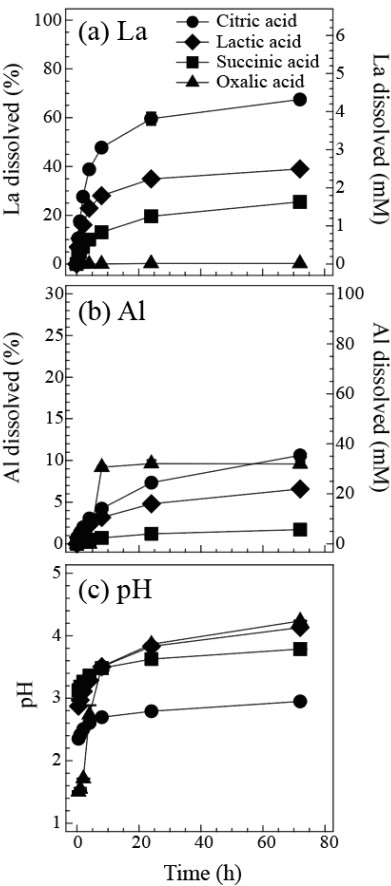

**Figure 3.** Changes in the total soluble concentrations of (**a**) La and (**b**) Al and in the (**c**) pH during the chemical leaching of the spent FCC catalysts using citric acid (●), lactic acid (◆), succinic acid (■) and oxalic acid (▲) at 50 mM.

Biogenic acids generally consist of a mixture of different types of organic acids. The results here indicate that the presence of oxalic acid in biogenic acid could significantly lower the overall dissolution of La. Therefore, the production of oxalic acid should be avoided during fungal fermentation.

Further chemical leaching tests using different citric acid concentrations resulted in final La dissolutions of 79%, 74%, 68%, 61% and 19% at 1 M, 100 mM, 50 mM, 30 mM and 10 mM citric acid, respectively (Figure 5a).

Since La-citrate complexes are formed with a molar ratio of 1:1 [19], a complete La dissolution from the spent FCC catalysts under these test conditions thoretically requires 6.7 mM citric acid. However, competition with other metal species (especially Al) necessitates an excessive amount of acid. The concentration of biogenic citric acid can range significantly, depending on the conditions and strains. Other studies reported that approximately 50–100 mM of citric acid (plus other organic acids) was produced by A. niger from sucrose and was used for spent FCC catalyst leaching [15,17]. Almousa et al. [23] compared several A. niger isolates and reported that citric acid production by the same strain could range between ~4 to 60 mM, depending on the medium and physicochemical conditions. Feeding local agro-wastes and byproducts resulted in further improvements of >1200 mM in citric acid production [23]. In this study, the effectiveness of 100 mM citric acid (within the microbiologically producible concentration range) was nearly comparable to that of 1 M citric acid (Figure 5a), despite a relatively large pH difference (Figure 5c). Overall, so far, the chemical leaching results predicted the potential usefulness of biogenic acids as a lixiviant for La leaching under the conditions that over ~100 mM of citric acid be concentrated during fermentation while suppressing oxalic acid formation.

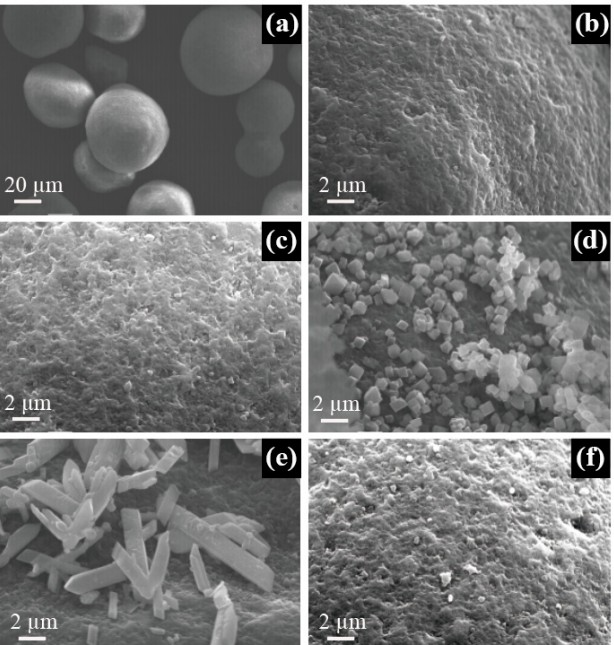

**Figure 4.** SEM images of the spent FCC catalyst surface (**a**,**b**) before and (**c**–**f**) after the chemical leaching test, using (**c**) citric acid, (**d**) succinic acid, (**e**) oxalic acid and (**f**) lactic acid at 50 mM.

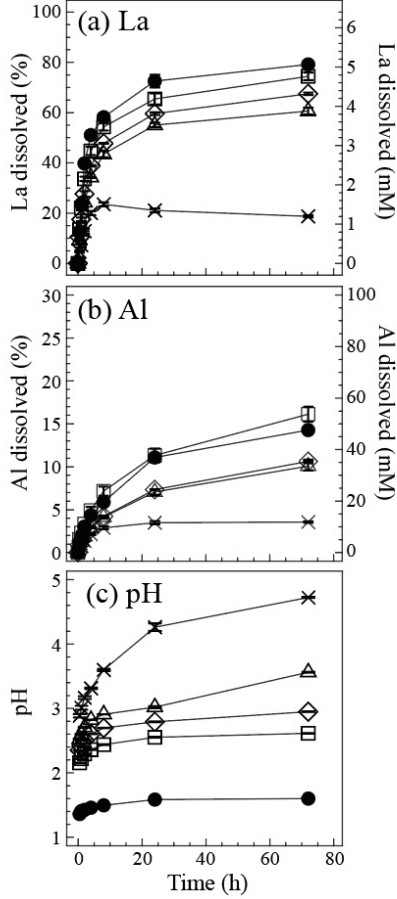

**Figure 5.** Changes in the total soluble concentrations of (**a**) La and (**b**) Al and (**c**) in the pH during the chemical leaching of the spent FCC catalysts using different citric acid concentrations (● 1M; □ 100 mM; ◇ 50 mM, △ 30 mM; × 10 mM).

### 2.3. Biogenic Acid Leaching of Spent FCC Catalysts

2.3.1. Biogenic Acid Production by Aspergillus Niger WU-2223L

In this study, the fungal fermentation step was separated from the leaching step, in order to avoid inhibitory effects of spent catalysis on fungal growth: In a recent study by Mouna and Baral [17], fermentation and leaching were simultaneously conducted in one step, wherein citric acid production was progressively inhibited at higher pulp densities (3–5%).

The chemical leaching results (Section 2.2) indicated that the concentration range of biogenic citric acid was potentially effective for La leaching from spent FCC catalysts. On the other hand, it was suggested that the fermentation of oxalic acid should be controlled in order to avoid the precipitation of La. To this end, biogenic acids were fermented by *A. niger* under different conditions. A 720 h fermentation using an initial spore density of $1 \times 10^6$ and $1 \times 10^7$ spores/mL resulted in a contrasting fungal morphology in the culture. Starting at $1 \times 10^6$ spores/mL produced a spherical pellet-like flocculation of fungal biomass (Figure 6a), whereas starting at $1 \times 10^7$ spores/mL exhibited a dispersed free filamentous form (Figure 6b).

**(a)**          **(b)**

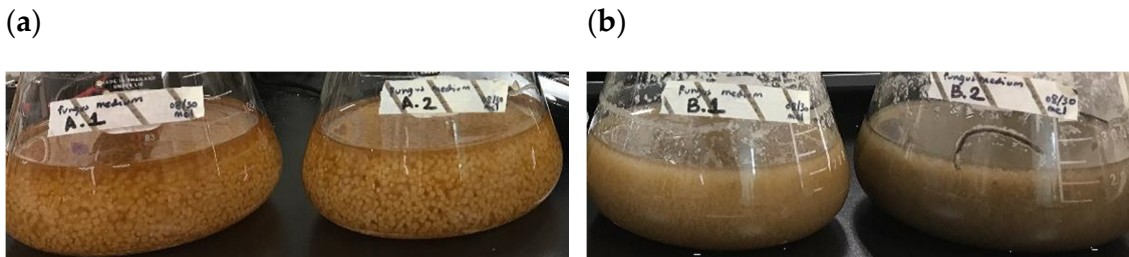

**Figure 6.** Morphological differences after a 720 h incubation of *A. niger* WU-2223L initiated at a spore density of (**a**) $1 \times 10^6$ spores/mL or (**b**) $1 \times 10^7$ spores/mL.

Considering the long study history (since the 1950s) of citric acid fermentation by *A. niger* strains, the effect of the spore inoculum level on fungal morphology is a somewhat neglected area. Papagianni and Mattey [24] described the relationship between the spore inoculum level and the morphology of *A. niger* in a submerged citric acid fermentation culture: Increasing the inoculum level from $10^4$ to $10^9$ spores/mL lowered the dissolved oxygen level, with a clear morphological transition from a pelleted to a dispersed mycelium form. It was suggested that controlling the mycelial morphology was of importance in ensuring an increased productivity. Veiter et al. [25] recently reviewed the relationship between the morphology of filamentous fungi and their productivity. The authors mentioned that such interlinkages are not yet entirely clear, as conflicting reports can also be found [25].

In this test, using an initial density of $1 \times 10^6$ spores/mL produced 140 mM citric acid after a one-month incubation. In contrast, the amount of citric acid production was negligible when starting at $1 \times 10^7$ spores/mL (Figure 7a). Instead, an ethanol odor was detected at 240 h from the latter flasks, which then increasingly became noticeable overtime during the 720 h incubation (by an olfactory observation). At 240 h, in fact, the production of ethanol was already confirmed by HPLC (Figure S2). The glucose consumption profile was relatively similar in both conditions (Figure 7a), as was the dry fungal biomass weight (Figure 7b). Other studies indeed confirmed the capability of ethanol fermentation by an *A. niger* strain [26]. Additionally, some *A. niger* strains were reported to grow at low oxygen tensions [27]. Therefore, the results obtained here implied that the different inoculum levels (thus, probably, different oxygen availabilities) triggered the alteration of the main metabolic pathway for the growth of *A. niger* WU-2223L, from aerobic citric acid fermentation to ethanol fermentation under oxygen limitation (Figure S3). The results here also suggested that the fungal morphology was strongly associated with the inoculum level (oxygen availability), as well as with the productivity of citric acid. From here on, therefore, the initial spore density of $1 \times 10^6$ was regularly used.

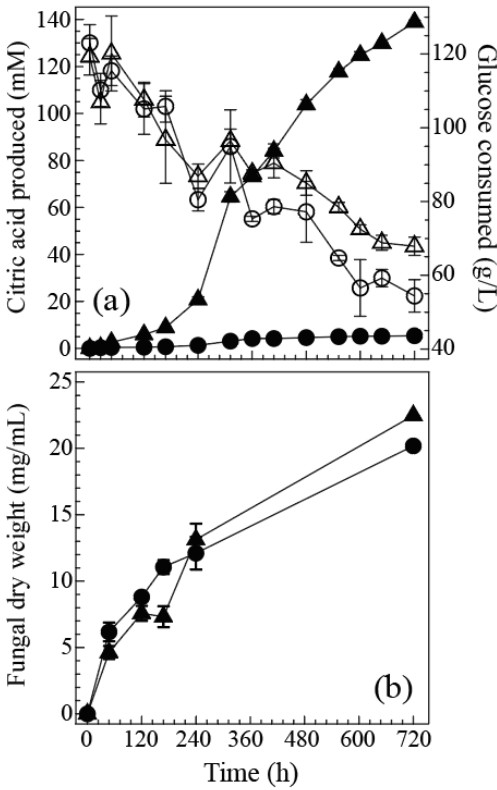

**Figure 7.** The effect of different inoculum levels of *A. niger* WU-2223L (▲,△ $1 \times 10^6$ spores/mL; •, ○ $1 \times 10^7$ spores/mL): (**a**) citric acid production (left axis; ▲, •) and glucose consumption (right axis; △, ○). (**b**) fungal dry weight.

The following tests employed three different conditions, and the corresponding results are shown in Table 2. Using a higher glucose dose did not necessarily result in a greater citric acid production: Halving the glucose concentration from 120 g/L (Condition A) to 60 g/L (Condition B) at $pH_{ini}$ 3.0 did not negatively affect the citric acid production (Table 2). Instead, unwanted oxalic acid was produced (31 mM) when excess glucose was used (Condition A). The production of oxalic acid in Condition A was likely caused via the oxaloacetate hydrolase (OAH)-catalyzed hydrolytic cleavage of oxaloacetate in the tricarboxylic acid (TCA) cycle [28]. The present results suggest that lowering the initial glucose concentration can reduce this by-reaction by repressing the OAH activity (Figure S3).

**Table 2.** Biogenic acids were produced after 340 h of submerged culture fermentation by *A. niger* WU-2223L under different conditions.

| Initial Conditions | A | B | C |
|---|---|---|---|
| Glucose | 120 g/L (670 mM) | 60 g/L (330 mM) | 120 g/L (670 mM) |
| $pH_{ini}$ | 3.0 | 3.0 | 2.0 |
| After 340 h fermentation | | | |
| Citric acid | 125 mM | 129 mM | 28 mM |
| Oxalic acid | 31 mM | n.d. [1] | n.d. |
| Lactic acid | n.d. | t.a. [2] | t.a. |
| Succinic acid | t.a. | t.a. | t.a. |
| Ethanol | n.d. | n.d. | t.a. |
| $pH_{fin}$ | 1.8 | 2.1 | 2.0 |
| Remaining glucose | 64 g/L (360 mM) | 24 g/L (130 mM) | 84 g/L (470 mM) |

[1] not detected by HPLC (Figure S4). [2] trace amount (<1 mM) detected by HPLC (Figure S4).

### 2.3.2. Biogenic Acid Leaching of Spent FCC Catalysts

Biogenic acids (A, B and C; Table 2) produced by *A. niger* under three different conditions (Section 2.3.1) were separated from the fungal biomass, and the cell-free spent media were used for the leaching test at a catalyst pulp density of 5%. In addition, by mimicking the chemical composition of the real biogenic acids after 340 h fermentation (A, B and C; Table 2), artificially reconstituted biogenic acids were prepared by mixing chemical reagents (Art-A, Art-B and Art-C), and also by mixing chemical reagents with glucose (Art-Ag, Art-Bg and Art-Cg), as shown in Table 3. The pH values indicated in Table 3 are the natural pH resulting from the mixed reagents.

**Table 3.** Artificially reconstituted biogenic acids.

| Artificially Reconstituted Biogenic Acids | Art-A | Art-B | Art-C | Art-Ag | Art-Bg | Art-Cg |
|---|---|---|---|---|---|---|
| Citric acid | 125 mM | 129 mM | 28 mM | 125 mM | 129 mM | 28 mM |
| Oxalic acid | 31 mM | - | - | 31 mM | - | - |
| Glucose | - | - | - | 64 g/L | 24 g/L | 84 g/L |
| pH | 1.5 | 1.9 | 2.3 | 1.5 | 1.9 | 2.3 |

The greatest effect was observed when the biogenic acids B and reconstituted biogenic acids Art-B and Art-Bg (129 mM citric acid) were applied, achieving a final La dissolution of ~74% (Figure 8a): The performance differences between these acids were only negligible, indicating that neither the origin of the acid (biotic or abiotic) nor the remaining glucose negatively affected the La leaching behavior. Although the biogenic acids A and reconstituted biogenic acids Art-A and Art-Ag contained an equivalent amount of citric acid (125 mM), La dissolution was significantly suppressed (<23%) due to the precipitation of La-oxalate, as was predicted in Section 2.1. The biogenic acids C and reconstituted biogenic acids Art-C and Art-Cg contained the lowest citric content of 28 mM, and 40–55% of La was shown to be dissolved (Figure 8a). The pH trend corresponded to the citric acid concentration in biogenic acids (Figure 8b). In some studies, biogenic leaching agents were reported to be more effective than the abiotically prepared counterparts due to the presence of unknown microbiological exudates [15,16]. However, under the conditions tested in this study, such an extra effect was not observed.

A recent study conducted by Mouna and Baral [17] reported advantages of one-step La bioleaching (simultaneous fermentation and leaching) from spent FCC catalysts at a lower pulp density of 1%, achieving a final La dissolution of 63% with a maximum of 98 mM biogenic citric acid (or ~80% when calcination was applied to the catalyst sample). However, increasing the pulp density (to 3% or 5%) in the one-step bioleaching inhibited fungal growth, suppressing the final La dissolution down to 52% or 33%, respectively [17]. The authors mentioned that the use of cell-free spent media significantly lowered the final La dissolution to 31% (at a 1% pulp density) [17]. Meanwhile, this study focused on investigating the utility of cell-free spent medium (the two-step process separating the initial fermentation step from the leaching step) in order to enable the treatment of higher catalyst pulp densities. The results of this study showed that the fungal fermentation of citric acid could be optimized while controlling the formation of unwanted oxalic acid. By directly applying the cell-free spent medium as biogenic acids, 74% of La was effectively dissolved from spent FCC catalysts at a pulp density of 5%. Overall, microbiological fermentation of organic acids could become a promising approach to supply an effective and environmentally benign lixiviant for REE scavenging from spent catalyst wastes.

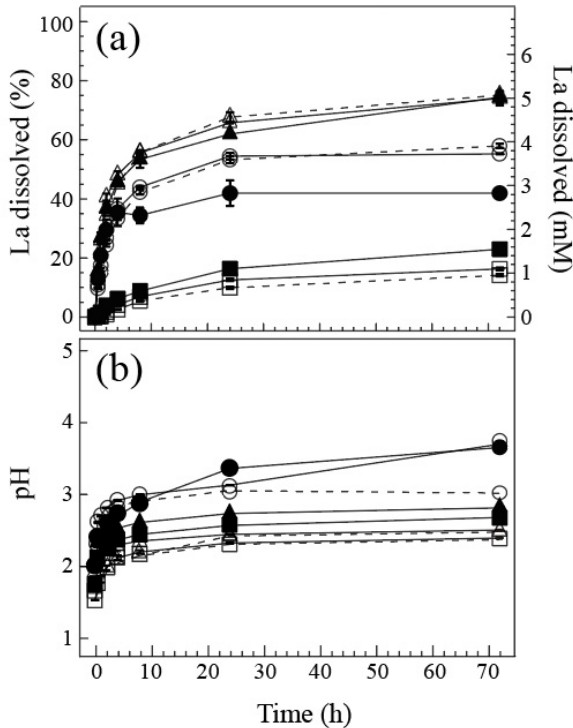

**Figure 8.** Dissolution of La from the spent FCC catalyst (at a pulp density of 5%) by biogenic acids (A:■, B:▲, C:●; Table 2) or artificially reconstituted biogenic acids without glucose (Art-A: □, Art-B: △, Art-C: ○ solid lines; Table 3) and with glucose (Art-Ag: □, Art-Bg: △, Art-Cg: ○ dotted lines; Table 3). Overtime changes in the (**a**) La concentrations and (**b**) pH are shown.

## 3. Materials and Methods

### 3.1. Spent FCC Catalyst Sample

The spent catalyst sample was provided by an Indonesian oil refining and distribution company (Refinery Unit VI Balongan, PT Pertamina (Persero), Jakarta, Indonesia). The catalysts were initially used in the FCC unit. The catalysts, deactivated by heavy metal contaminants (e.g., Ni, V) deriving from crude oil, were subjected to a regeneration treatment under a high pressure and temperature to trap toxic metals inside the zeolite structure. After repeating the usage/regeneration cycles several times, the FCC catalysts were transferred and further utilized in the residual fluid catalytic cracking (RFCC) unit to process heavier crude oil. After repeated usage/regeneration cycles in the RFCC unit, the catalyst came to its end-of-cycle. The spent catalyst sample used in this study was calcined (at 725 °C, 1.4 kg/cm$^2$) for 10 min to remove impurities such as coke and oil on the surface.

### 3.2. Characterization of Spent FCC Catalyst

#### 3.2.1. X-ray Diffraction (XRD)

The spent FCC catalyst sample was analyzed by XRD (Ultra IV; Rigaku, Tokyo, Japan; Cu K$\alpha$ 40 mV, 40 kV), before and after two different hydrogen fluoride (HF) treatments, as follows: 10 g of spent catalyst sample was mixed with 20 mL of 23% HF. In one condition, the mixture was placed in a 50 mL plastic vial and shaken at room temperature for 1 h. In the other condition, the mixture was placed in a Teflon vessel and microwaved (at 210 °C, 1000 W) for 50 min. After each treatment, the catalyst residue was collected by centrifugation, washed three times with 10 mL of deionized water, dried in an oven at 55 °C overnight, and analyzed by XRD. The HF treatments were applied in order to visualize the XRD patterns of minor components by partially digesting major zeolite-Y and Al$_2$O$_3$ peaks.

### 3.2.2. Sequential Acid Digestion

Step 1: Dry spent catalyst powder (2.5 g) was mixed with 20 mL of 1 M HCl in a 50 mL plastic vial and shaken for 16 h at 30 °C. After centrifugation, the leachate was collected, while the catalyst residue was washed with 10 mL deionized water three times. The leachate and all washing waters from Step 1 were pooled (Fraction 1) to measure metal concentrations by ICP-OES.

Step 2: The remaining residue from Step 1 was mixed with 30 mL of 46% HF and shaken for 1 h at 30 °C. After centrifugation, the leachate was collected, while the remaining residue was mixed with 30 mL of 46% HF and shaken for 16 h at 30 °C. Then, 5 g of $H_3BO_3$ was further added to the mixture and shaken for another 8 h at 30 °C. The leachate was collected, and the catalyst residue was washed with 10 mL of boiling deionized water three times. All leachates and washing waters from Step 2 were pooled (Fraction 2) to measure metal concentrations by ICP-OES.

Step 3: The remaining residue from Step 2 was mixed with 10 mL of concentrated $HNO_3$ and shaken for 2 h at 30 °C. After centrifugation, the leachate was collected, while the residue was washed with 15 mL of boiling deionized water once. The leachate and washing water from Step 3 were pooled (Fraction 3) to measure metal concentrations by ICP-OES. After this step, the spent catalysts were dissolved entirely, and no residues were visible. The metal composition of the spent catalyst was calculated by combining the results obtained from each fraction (Fractions 1, 2, 3). All tests were done in duplicates.

Generally, components that are solubilized by HCl (Step 1), HF (Step 2) or $HNO_3$ (Step 3) account for salt-type minerals, silicate group minerals or sulfides and their related minerals, respectively [29].

### 3.2.3. Morphology and Particle size

The surface morphology was investigated by sputter-coating the spent FCC sample with Au (MPS-1S; Vacuum Device Inc., Tokyo, Japan), followed by observation with a scanning electron microscope (SEM; VE-9800; Keyence, Osaka, Japan) at an acceleration voltage of 20 kV and at a ×500 magnification. The particle size of the spent catalyst was analyzed by a particle size distribution analyzer (LA-950; Horiba, Tokyo, Japan).

### 3.3. Chemical Leaching of Spent FCC Catalyst

All chemical leaching tests were done in 300-mL Erlenmeyer flasks containing 100 mL of acid lixiviant at a pulp density of 5% (*w/v*). Flasks were incubated and shaken at 120 rpm and at 30 °C. The first test compared 1M HCl, 1M $H_2SO_4$ and 1 M citric acid. The second test compared different concentrations of citric acid (1 M, 100 mM, 50 mM, 30 mM or 10 mM). The last test compared different organic acids at 50 mM (citric acid, oxalic acid, succinic acid or lactic acid). All tests were done in duplicate flasks. Samples were taken periodically to measure the pH and metal concentrations by ICP-OES (Perkin Elmer Optima 8300).

### 3.4. Biogenic-Acid Leaching of Spent FCC Catalyst

### 3.4.1. Biogenic Acid Production

Effect of initial spore density: A filamentous fungal species, *Aspergillus niger* WU-2223L (known as a hyperproducer of citric acid [30]), was purchased from NITE Biological Resource Center (NBRC; Tokyo, Japan), and was used in this study. Fungal spores were cultivated on a slant of 4% (*w/v*) potato dextrose agar medium at room temperature (25–30 °C) for about 5–7 d. Spores were then washed with 9 g/L NaCl saline solution and inoculated at two different densities ($1 \times 10^6$ or $1 \times 10^7$ spores/mL) into 250 mL basal salts medium (3 g/L $(NH_4)_2SO_4$, 1 g/L $K_2HPO_4$, 1 g/L $KH_2PO_4$, 500 mg/L $MgSO_4 \cdot 7H_2O$, 10 mg/L $FeCl_3 \cdot 6H_2O$, 14 mg/L $MnSO_4$; pH 3.0 with $H_2SO_4$; [31]) containing 2% (*v/v*) methanol and 120 g/L glucose (in 500-mL Erlenmeyer flasks). The citric acid production by *A. niger* WU-2223L was reported to be optimal under this acidic pH in the presence of methanol [30,32]. The flasks were incubated and shaken at 120 rpm and at 30 °C for 720 h.

Effect of substrate concentration and pH: The initial spore density was fixed at $1 \times 10^6$ spores/mL, and spores were inoculated into 200 mL of the same basal salts medium containing 2% (*v/v*) methanol and two different concentrations of glucose (120 g/L or 60 g/L) set at two different initial pHs (2.0 or 3.0 with with $H_2SO_4$) (Table 2). The flasks were incubated and shaken at 120 rpm and at 30 °C for 340 h.

Glucose concentrations were measured by the mutarotase-GOD method (LabAssay™ Glucose; Wako Pure Chemical Industires, Osaka, Japan). The culture supernatant was analyzed for the organic acid composition by HPLC (CO-2065 Plus; Jasco, Tokyo, Japan; column temperature 60 °C using 2 mM $HClO_4$ as a mobile phase at 0.7 mL/min.) Citric acid concentrations were measured colorimetrically by the pyridine-acetic anhydride method [33]. Oxalic acid concentrations were calculated by quantitative analysis of the HPLC results.

### 3.4.2. Biogenic Acid Leaching

Biogenic acid produced by *A. niger* (Section 3.4.1) was collected (as spent media) by filtration (0.2 μm) to remove all fungal biomass from the culture. Spent catalysts were added (at a pulp density of 5% (*w/v*)) into 100 mL of this biogenic acid in 300-mL Erlenmeyer flasks. The flasks were incubated and shaken at 120 rpm and at 30 °C. Liquid samples were periodically withdrawn with micropipettes to measure the pH and metal concentrations. All tests were done in duplicate flasks. Control tests were set up to compare the effectiveness of real biogenic acids to artificially reconstituted biogenic acids (mixtures of chemical reagents). All tests were done in duplicate flasks.

All chemical reagents used in this study were analytical grade and manufactured by Wako Pure Chemical Industries, Osaka, Japan.

## 4. Conclusions

The spent FCC catalyst consisted of zeolite-Y/$Al_2O_3$ containing La (1.9%) and Ni (1.2%; originating from crude oil) in the form of $La_2O_3$ and $Ni_2SiO_4$, respectively.

Based on the chemical leaching tests, the performance of citric acid (>100 mM) was nearly comparable to that of conventional inorganic acids (1 M), owing to the dual effect of La-citrate complex formation and acidolysis.

The initial conditions for biogenic acid fermentation by *A. niger* (e.g., inoculum level, substrate concentration, pH) largely affected the resultant biogenic acid composition, and its manipulation was possible in order to concentrate preferable citric acid (~130 mM) while suppressing the production of unwanted oxalic acid.

The effectiveness of actual biogenic acid was nearly identical to that of artificially reconstituted biogenic acids. A final La dissolution of ~74% was achieved in both cases.

Fungal fermentation of organic acids is a promising, sustainable approach to supplying an effective and environmentally benign alternative lixiviant for REE scavenging from spent FCC catalyst wastes. The schematic flowsheet of this process is summarized in Figure 9.

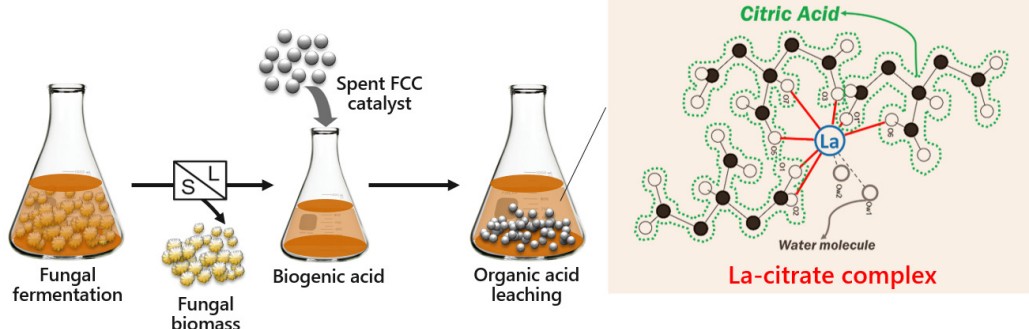

**Figure 9.** Schematic flowsheet of the biogenic acid leaching of REE from spent FCC catalysts. The structure of the La-citrate complex was adopted and modified from Vanhoyland et al. [19].

**Supplementary Materials:** The following are available online at http://www.mdpi.com/2073-4344/10/9/1090/s1. Figure S1: Changes in total soluble concentrations of La (a), Al (b) and Ni (c) during chemical leaching of spent FCC catalysts using 2 M $H_2SO_4$ (■) or 2 M HCl (▲). Figure S2: HPLC result of biogenic acids produced by *A. niger* WU-2223L (2-times dilution; at 240 h) under the initial conditions of; $1 \times 10^7$ spores/mL, 120 g/L glucose; pH 3.0. Compared to Figure S4a (initial spore density of $1 \times 10^6$ spores/mL was used instead of $1 \times 10^7$ spores/mL), production of ethanol was confirmed while that of citric acid was greatly suppressed. Figure S3: Metabolic reaction network of *A. niger* for fermentation of citric acid, oxalic acid and ethanol from glucose. (adapted and modified from [34].) The addition of methanol in fungal media was reported to increase the citric acid production via repression of 2-oxoglutarate dehydrogenase [31]. Figure S4: HPLC results of biogenic acids produced by *A. niger* WU-2223L (without dilution; at 340 h) under different initial conditions; (a) glucose 120 g/L, pH 3.0, (b) glucose 60 g/L, pH 3.0, (c) glucose 120 g/L, pH 2.0. The initial spore density was set to $1 \times 10^7$ spores/mL in all cases.

**Author Contributions:** Conceptualization, N.O. and H.T.B.M.P.; Methodology, M.P.D. and N.O.; Validation, N.O.; Formal analysis, M.P.D.; Investigation, M.P.D.; Resources, N.O.; Writing—original draft preparation, M.P.D.; Writing—review and editing, N.O.; Visualization, N.O.; Supervision, N.O.; Project administration, N.O. and H.T.B.M.P.; Funding acquisition, N.O. All authors have read and agreed to the published version of the manuscript.

**Funding:** This research was partly funded by JSPS (Japan Society for the Promotion of Science) KAKENHI Grant Number JP20H00647.

**Acknowledgments:** M.P.D. is grateful for the Indonesia Endowment Fund for Education (LPDP scholarthip). We thank PT. Pertamina (Persero), Indonesia for kindly supplying spent FCC catalyst samples.

**Conflicts of Interest:** The authors declare no conflict of interest.

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
