# Peer review of "Recovering Secondary REE Value from Spent Oil Refinery Catalysts Using Biogenic Organic Acids"

_catalysts, doi:10.3390/catal10091090_

Round 1

Author Response

separate file attached.

Reviewer 2 Report

Dear Editor

There are the recommendation and comments for this paper “Recovering secondary REE value from spent oil refinery catalysts using biogenic organic acids”.

Recommendation: Minor Revision

This study aimed to exploit the leachability of spent FCC catalysts as a secondary source for La, by using an alternative organic acid lixiviant produced under optimized fungal fermentation condition. Microbiological fermentation of organic acids could become a promising approach to supply efficient and environmentally-benign alternative lixiviant for REE scavenging from spent FCC catalyst wastes.

I think that the research is studied well. But there are only several minor-points to be revised as follows.

Comments 1:  I think that the author should supply the commercial information of reagents they used in this manuscript, eg (NH4)2SO4, K2HPO4, KH2PO4, MgSO4.7H2O, FeCl3.6H2O, MnSO4 and so on.

Comment 2:  It is easy to understand the experiments procedure of this paper, if the author can draw a flow chart or scheme for the recovering secondary REE value from spent oil refinery catalysts by biogenic organic acids.

Comment 3: P7 L219 “The authors mentioned that such interlinkages are not yet entirely clear, as conflicting reports can also be found.” The author should add reference to here.

Comment 4: “SEM images in Fig. 4 revealed the formation of distinctive precipitates on

the surface of spent FCC catalysts after the leaching test, especially in the case of oxalic acid (Fig. 4e) and succinic acid (Fig. 4d).” I think that the author can also do EDX analysis mapping profiles (point scan) showing selective elements, which also can support the results of the SEM images. The metal elements distribution is different in the formation of distinctive precipitates.

Comment 5: In Figure 7, it shows that the citric acid produced and time is days. In Figure 8, it shows the dissolution of La from spent FCC catalyst and time is hours. I suggest the author supply the citric acid produced by hours and time is same to the time of the dissolution of La from spent FCC catalyst.

Author Response

separate file attached.

Reviewer 3 Report

The manuscript “Recovering secondary REE value from spent oil refinery catalysts using biogenic organic acids” by Melisa Pramesti Dewi1, Himawan Tri Bayu, Murti Petrus, and Naoko Okibe* describes a method to recover the rare earth element lanthanum, used as a catalyst in combination with other substances in the oil refining industries, after use (the spent catalyst) and when it is no longer suitable for further recycling. In this particular study the spent catalyst used in crude oil refining consisted of zeolite-Y/Al2O3 containing La (1.9%) and Ni (1.2%) in the form of La2O3 and Ni2SiO4, respectivel.The main feature of the method is the use of acids to extract La from waste materials using this catalyst as an example. However, instead of using mixtures of mineral acids, which extract more than one metal unselectively, e.g. Al, Si, La, Ni, Fe, the authors developed a procedure in which acids produced in culture by certain fungi (biogenic acids) could be used instead. This work showed that cultures of the fungus A. Niger WU-2223L, grown under special conditions determined during the course of this work (e.g., inoculum level, substrate concentration, pH), produced mostly citric acid during fermentation, and this acid caused the dissolution of La from the spent metal and excluded other metals.It was shown that the effectiveness of the biogenic acid was nearly identical to that of artificial samples of biogenic acids prepared by mixing chemicals from a bottle. The final La dissolution of 74% was achieved in both cases.

This study showed that microbiological fermentation of organic acids is a promising approach to produce an effective, low cost and environmentally-benign lixiviant for rare earth element scavenging, e.g. La as in this work, from spent catalyst wastes.

There have not been many similar works.In a related recent study (ref. 14) also showing the advantages of bioleaching, a lower concentration of La could be obtained (63%). A higher concentration (80%) could be obtained if calcination was used.

The work presented in this study is interesting and well described, and it should be of interest to the readers of Catalyst, therefore I recommend it for publication, once a few minor corrections, listed below, have been attended to.

Minor corrections required:

Page 1, line 37: it is not necessary to use the symbol for lanthanum in brackets. It is not an abbreviation but a chemical symbol. The same apples to cerium (Ce) in line 63.

Results and Discussion:

Table 1. It would be better if this Table had a footnote stating that the resutts obtained are the average of duplicate determinations each.

Line 138: remove the semi-colon after molecules.

Line 212: “of fungal morphology” => “on fungal morphology”

Line 217: “The linkage” => “the relationship”

Line 225: “These results suggest that the different inoculum level (thus different oxygen availability) triggered alteration of the main metabolic pathway for fungal growth”. This may be the reason yes, but it was not determined to be so for sure (i.e.less oxygen availability).Was there really so much less oxygen available when 10x more fungus was used?

It would be better that the authors do not state this as if it were a proven fact, but rather as a probability, i.e. ““These results suggest that the different inoculum level (thus probably different oxygen availability) triggered alteration of the main metabolic pathway for fungal growth”.

This comment applies to line 229.

Table 2: The footnotes should be numbered.

The results described in section 2.3.2 need a few extra words to be clear. When I first read the text it was not clear which were the biogenic acid solutions and which were solutions prepared by mixing reagents and what happened to them but only after reading through the entire section. This arises from the fact that the graph in Figure 8 is a combination of results described throughout this section. I suggest including a few words in some places in the text to avoid confusion and clarify matters. There is no need to underline them, I did so merely to show better what I mean:

Line 253: “Biogenic acids produced by A. niger under three different conditions(section 2.3.1)” will be better as “Biogenic acids (A, B, and C) produced by A. niger under three different conditions (section 2.3.1),…”

Line 254: ---“and the cell-free spend media were used for the leaching test at a

catalyst pulp density of 5%. By mimicking the chemical composition….” This sentence should be followed by a comment like the one shown: “---“and the cell-free spend media were used for the leaching test at a catalyst pulp density of 5%. The results are shown in Figure 8. In addition, by mimicking the chemical composition…..”

Line 255: “By mimicking the chemical composition of the real biogenic acids (Table 2)”. This statement will become clearer if it is stated instead “….In addition, by mimicking the chemical composition of the real biogenic acids according to the results obtained (Table 2)….”.

Line 256: “artificially-reconstituted biogenic acids were prepared by mixing chemical reagents” would be better as “artificially-reconstituted biogenic acids (Art-A to Art-C) were prepared by mixing chemical reagents, and also by mixing chemical reagents with glucose (Art-Ag to Art-Ac)”

Table 3 does not provide any new information on its own. In fact it could even have been in the experimental section. The best would be to keep this table attached to figure 8. For example, the graph on top and the table with the composition of each solution tested (the so-called artificially-reconstituted biogenic acids) immediately below, so that the reader knows what each curve in the figure represents.

The authors should read through the manuscript and correct minor language errors which occur.

Author Response

separate file attached.

Reviewer 4 Report

After reading the manuscript, I regard it as an interesting work worthy of publication in this journal. Before doing so, I have a couple of questions to ask. Firstly, what does it happen with V? this is usually another important contaminant of FCC catalyst and it is scarcely mentioned in the paper.Secondly, the authors should clarify in the text that the biogenic acids used stem from the fermentation, but these acids are separated from the fungus prior to its use. This is a bit messy in the text and the authors should state it more clearly.

Author Response

separate file attached.

Round 2

Reviewer 1 Report

The authors have improved the discussion and the methods description, which I believe will benefit the readers. Unfortunately, they are unwilling or unable to any of the experiments suggested, which could improve the attractiveness of the article.

I have a couple minor comments that the authors may want to address before publication.

-The following publication may be interesting for the introduction, as it illustrates importance of La in cracking catalysts and its limited supply to global markets:
https://doi.org/10.1021/acscatal.7b02011

-Please revise the y-axis limits in the figures so that trends can be observed (e.g. Figure 2c, Fig. 3b).

-I am suprised that an FCC catalyst can be imaged at 20 keV without charging.

-I would double check that all the data previously stated as "not shown" is now included.

Author Response

a separate file attached
